# Synaptic transmission and plasticity require AMPA receptor anchoring via its N-terminal domain

**Jake F Watson, Hinze Ho, Ingo H Greger\***

Neurobiology Division, MRC Laboratory of Molecular Biology, Cambridge, United Kingdom

**Abstract** AMPA-type glutamate receptors (AMPARs) mediate fast excitatory neurotransmission and are selectively recruited during activity-dependent plasticity to increase synaptic strength. A prerequisite for faithful signal transmission is the positioning and clustering of AMPARs at postsynaptic sites. The mechanisms underlying this positioning have largely been ascribed to the receptor cytoplasmic C-termini and to AMPAR-associated auxiliary subunits, both interacting with the postsynaptic scaffold. Here, using mouse organotypic hippocampal slices, we show that the extracellular AMPAR N-terminal domain (NTD), which projects midway into the synaptic cleft, plays a fundamental role in this process. This highly sequence-diverse domain mediates synaptic anchoring in a subunit-selective manner. Receptors lacking the NTD exhibit increased mobility in synapses, depress synaptic transmission and are unable to sustain long-term potentiation (LTP). Thus, synaptic transmission and the expression of LTP are dependent upon an AMPAR anchoring mechanism that is driven by the NTD.

## Introduction

AMPA receptors (AMPARs) are embedded at postsynaptic sites, aligned with the presynaptic glutamate release machinery for optimal signaling (*Lisman et al., 2007*). Their activation drives propagation of presynaptic impulses through depolarization of the postsynaptic membrane (*Traynelis et al., 2010*). As AMPARs have low apparent glutamate affinity, and rapidly diffuse in the plane of the membrane, they require trapping at synaptic sites in order to effectively contribute to signal transmission (*Choquet and Triller, 2013*; *Heine et al., 2008*). Synapse strengthening, as occurs during learning, results from the recruitment of additional AMPARs and their enrichment at synapses (*Chater and Goda, 2014*; *Huganir and Nicoll, 2013*; *Kessels and Malinow, 2009*). Hence, the mechanisms underlying AMPAR positioning are fundamental to synaptic transmission and plasticity.

Signaling properties and synaptic delivery depend on AMPAR composition. AMPARs are tetramers, assembled from the core GluA1-GluA4 (pore-forming) subunits, which associate with a varying set of auxiliary subunits, such as transmembrane AMPAR regulatory proteins (TARPs) (*Jackson and Nicoll, 2011*). Each core subunit consists of four domains – a short cytosolic C-terminus (CTD), the transmembrane ion channel domain (TMD), and two extracellular domains: the ligand-binding domain (LBD) and the distal N-terminal domain (NTD) (*Figure 1A*).

The sequence-diverse C-termini mediate subtype-selective AMPAR trafficking, and their role in recruitment of specific subunits during synaptic plasticity has been extensively studied (*Derkach et al., 2007*; *Kessels and Malinow, 2009*; *Newpher and Ehlers, 2008*; *Shepherd and Huganir, 2007*). The CTD interacts with postsynaptic scaffolding proteins (*Anggono and Huganir, 2012*; *Shepherd and Huganir, 2007*), but deletion of this region is not a prerequisite for receptor clustering (*Bats et al., 2007*; *MacGillavry et al., 2013*), and how critical these interactions are in

**\*For correspondence:** ig@mrc-lmb.cam.ac.uk

**Competing interests:** The authors declare that no competing interests exist.

**eLife digest** Neurons send signals via electrical impulses that are transmitted between cells by small molecules known as neurotransmitters. The information is passed from neuron to neuron at specialized points of contact termed synapses. On release of neurotransmitters from the first neuron, the molecules attach to 'docking stations' called receptors on the next neuron, referred to as the postsynaptic cell.

One of these receptors, the AMPA receptor, transmits signals by binding to a neurotransmitter called glutamate. Previous research has shown that in order to bind glutamate effectively, these receptors need to be trapped and anchored at the correct location at the synapse. This trapping mechanism controls the number of receptors present, which strengthens the synapse, and ultimately mediates learning and memory. However, it is still not clear how AMPA receptor trapping is achieved.

To investigate this question, Watson et al. examined how AMPA receptors (and mutant forms of the receptor) affect the communication between neurons using brain slices from mice. The experiments show that an external segment of the AMPA receptor called the N-terminal domain (or NTD for short) is a key element for receptor anchoring at the postsynapse. The AMPA receptor is made out of four different subunits; when the NTD portion was removed from one specific subunit, fewer receptors were anchored correctly at the postsynapse. When the NTD was removed from another subunit, it completely prevented the synapse from learning. Therefore, the NTD brings about subunit-selective anchoring of the AMPA receptor, which affects the ability of the synapse to transmit signals.

Important next steps would be to identify the proteins that interact with the NTD and how this specific anchoring affects the strength of the synapse. Another key step will be to understand what mechanisms control the number of AMPA receptors at synapses, to ultimately enable learning.

plasticity is not fully understood (*Boehm et al., 2006*; *Granger et al., 2013*; *Kim et al., 2005*). Currently, the best-described anchoring mechanism is mediated by TARP $\gamma-2$, which interacts via its C-terminus with the scaffolding protein PSD-95, and limits diffusion of synaptic AMPARs (*Opazo et al., 2012*; *Schnell et al., 2002*).

Like the CTD, the NTD is highly sequence-diverse between the four AMPAR subunits, offering great capacity for subunit-specific control. This domain projects into the crowded environment of the synaptic cleft, providing a large, structurally dynamic docking platform (*García-Nafría et al., 2016a*). For example, neuronal pentraxins (NPs) interact with the AMPAR NTD and mediate clustering at interneuron synapses, but the underlying mechanism remains to be clarified (*Chang et al., 2010*; *O'Brien et al., 1999*; *Sia et al., 2007*). Here we show that synaptic delivery of GluA1 and GluA2, prominent AMPAR subunits in CA1 pyramidal neurons (*Lu et al., 2009*), is dependent on their NTDs in a subunit-specific manner. Although receptors lacking the NTD accumulate at the extra-synaptic surface, they cannot effectively contribute to synaptic transmission and are unable to sustain LTP.

## Results

Although the NTD encompasses ~50% of an AMPAR subunit, its function beyond receptor assembly (*Herguedas et al., 2013*) is unclear. To study the role of this domain at synapses we expressed AMPAR subunits in organotypic hippocampal slices using single-cell electroporation of plasmid DNA (*Figure 1B*). Exogenously expressed AMPARs mostly form homomers (*Shi et al., 2001*), and when unedited at the Q/R site give rise to a rectifying current/voltage (I/V) relationship, resulting from intracellular polyamine block (*Bowie and Mayer, 1995*; *Kamboj et al., 1995*). Homomeric GluA1, and GluA2 unedited at the Q/R RNA editing site (denoted GluA2Q), are selectively blocked at positive membrane potentials, permitting electrophysiological detection of exogenous AMPARs (*Hayashi et al., 2000*). Untransfected neurons exhibit a linear I/V response resulting from endogenous heteromers containing the edited GluA2 subunit (GluA2R), and therefore the ratio of currents at positive and negative membrane potentials, the rectification index (RI), can be used to assay the

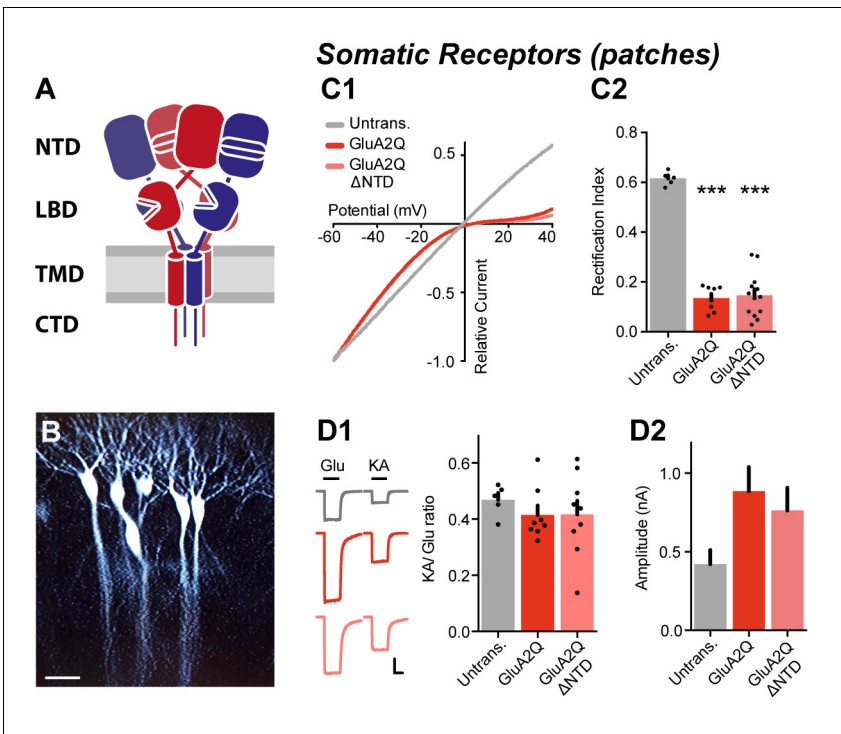

**Figure 1.** NTD deleted GluA2 is robustly expressed on the cell surface. (**A**) AMPA receptor schematic detailing the four-domain structure (NTD - N-terminal domain; LBD - Ligand-binding domain; TMD - Transmembrane domain; CTD - C-terminal domain). (**B**) Single-cell electroporated CA1 pyramidal neurons in an organotypic slice culture. Scale bar = 50 μm. (**C1**) I/V curves of glutamate-evoked AMPAR currents recorded from outside-out patches of untransfected, GluA2Q and GluA2Q ΔNTD-expressing cells. (**C2**) AMPAR currents from transfected neurons show strong inward-rectification on GluA2 construct expression (Rectification index (RI): untrans.: 0.62 ± 0.03 (n = 5); GluA2Q: 0.13 ± 0.02 (n = 8); GluA2Q ΔNTD: 0.15 ± 0.02 (n = 13); One-way ANOVA, p<0.0001). Significance (*) indicates difference to untransfected cells. (**D1**) Ratio of response amplitude to kainic acid and glutamate, indicative of auxiliary subunit association, from somatic patches is unchanged on receptor overexpression (KA/Glu: untrans.: 0.48 ± 0.03 (n = 5); GluA2Q: 0.41 ± 0.03 (n = 8); GluA2Q ΔNTD: 0.42 ± 0.05 (n = 9); One-way ANOVA, p=0.51). Example traces showing glutamate (Glu) and kainic acid (KA) application are shown left. Scale bar = 50 ms and 100 pA. (**D2**) Amplitudes of surface patch AMPAR glutamate responses are apparently elevated on GluA2Q or GluA2Q ΔNTD overexpression (untrans.: 398 ± 50 pA; GluA2Q: 886 ± 153 pA; GluA2Q ΔNTD: 763 ± 145 pA).

The following figure supplement is available for figure 1:

**Figure supplement 1.** NTD deletion construct screening.

synaptic receptor content. We have characterized synaptic responses from untransfected and transfected cells using two parameters: excitatory postsynaptic current (EPSC) amplitude, which is directly proportional to the number of synaptic receptors, and the RI, which provides a readout of the proportion of exogenous to endogenous receptors contributing to the response.

## GluA2 lacking the NTD is delivered to synapses but depresses synaptic transmission

We initially focused on the NTD of GluA2, a subunit most commonly incorporated into AMPARs (*Isaac et al., 2007*). To compare GluA2 wild-type (WT) to a mutant lacking the NTD, we first tested the surface trafficking capacity of GluA2 constructs bearing an NTD deletion, a modification that does not impair AMPAR function (*Pasternack et al., 2002*). Deletion of amino acids 1–377 of the mature polypeptide resulted in optimal surface expression in HEK293T cells (*Figure 1—figure supplement 1A–B2*) and was used throughout this study (designated GluA2Q ΔNTD). When expressed

in CA1 pyramidal neurons, this GluA2Q ΔNTD construct produced inwardly rectifying responses in somatic outside-out patches, matching the responses from neurons expressing full-length GluA2Q (*Figure 1C*). As the functional properties of neuronal AMPARs and their expression at synapses are modulated by auxiliary subunits, most prominently by TARPs, we determined whether exogenous GluA2 was still TARP-associated. A signature of TARP action is an increased efficacy of the partial agonist kainic acid (KA) (*Tomita et al., 2005*; *Turetsky et al., 2005*), and the ratio of kainate and glutamate response amplitudes (KA/Glu ratio) is a measure of AMPAR/TARP stoichiometry (*Shi et al., 2009*). The KA/Glu ratio suggests full TARP occupancy for both GluA2Q and GluA2Q ΔNTD (*Figure 1D1*), implying that TARPs are not limiting in our system and NTD deletion does not affect TARP association. Amplitudes of glutamate-evoked currents from GluA2Q and GluA2Q ΔNTD-expressing somatic patches were similar, further confirming that expression levels are comparable, and were approximately double that of untransfected neurons (*Figure 1D2*), which can be explained by the greater single channel conductance of (unedited) GluA2Q homomers than native receptors (see below; *Swanson et al., 1997*).

To assay synaptic responses, we stimulated Schaffer collateral fibers and simultaneously recorded whole-cell AMPAR responses from pairs of transfected and untransfected neurons. As observed for somatic receptors, both GluA2Q and GluA2Q ΔNTD-expressing cells produced strongly rectifying responses relative to untransfected neurons (*Figure 2A*), demonstrating that GluA2Q homomers lacking the NTD reach synapses. However, while EPSC amplitudes were elevated in neurons expressing GluA2Q relative to paired untransfected cells (166 ± 13%), EPSCs were significantly reduced in GluA2Q ΔNTD neurons (58 ± 5%; *Figure 2B1–2* versus *Figure 2C1–2*). This effect was specific to the synaptic AMPAR component, as NMDAR EPSCs were unchanged in both conditions (*Figure 2B3 and C3*). Therefore, GluA2Q ΔNTD receptors reach synapses but interfere with synaptic transmission.

## GluA2 ΔNTD reduces spontaneous transmission

As a change in AMPAR EPSC amplitude could occur for a variety of reasons, we sought to identify the mechanism for this effect. Paired-pulse ratios were unchanged in cells expressing either GluA2 construct and were comparable to untransfected cells (*Figure 2—figure supplement 1A*), suggesting a postsynaptic locus for the effect. As the GluA2 NTD has been implicated in spine formation (*Passafaro et al., 2003*; *Saglietti et al., 2007*), we assayed spine density, which was unchanged between the three conditions (GluA2Q, GluA2Q ΔNTD and untransfected; *Figure 2—figure supplement 1B*). Since NMDAR EPSCs were also unaffected in these neurons, a change in the number of synapses cannot explain this effect.

To characterize the postsynaptic response in greater detail we recorded AMPAR miniature EPSCs (mEPSCs). In line with the changes in evoked transmission, spontaneous transmission was dramatically impaired on NTD deletion. mEPSC amplitudes of GluA2Q ΔNTD cells were significantly reduced relative to GluA2Q-expressing cells (*Figure 3A*), while decay kinetics was unaffected by NTD deletion (*Figure 3C*). We also noted a highly significant decrease in mEPSC frequency (*Figure 3B*) between GluA2Q and GluA2Q ΔNTD-expressing cells. While an increase in mEPSC amplitude on GluA2Q expression (as per evoked EPSCs) was not observable, an increase in mEPSC frequency was apparent, which could be caused by an increase in mEPSC amplitude, resulting in small events emerging from below the detection limit (see Materials and methods).

To determine whether the depression of EPSC amplitudes could be explained by reduced single-channel conductance, caused by deleting the NTD, we conducted non-stationary fluctuation analysis (NSFA) from mEPSCs (*Figure 3D*). The synaptic AMPAR single-channel conductance was significantly increased in both groups of transfected cells, as expected from the expression of Q/R-unedited receptors (*Swanson et al., 1997*). However, there was no difference between GluA2Q and GluA2Q ΔNTD that could explain the substantial drop in synaptic AMPAR current amplitudes in GluA2Q ΔNTD-expressing neurons (*Figure 2C*). Based on this accumulated evidence, a change in the number of receptors at the synapse most feasibly explains the observed effect on EPSC amplitude, whereby significantly less GluA2Q ΔNTD receptors are present at the synapse than GluA2Q. Since AMPARs have a relatively low affinity for L-glutamate (e.g. *Jonas, 2000*), the NTD may stabilize and cluster the receptor in proximity to presynaptic release sites to enable optimal receptor activation.

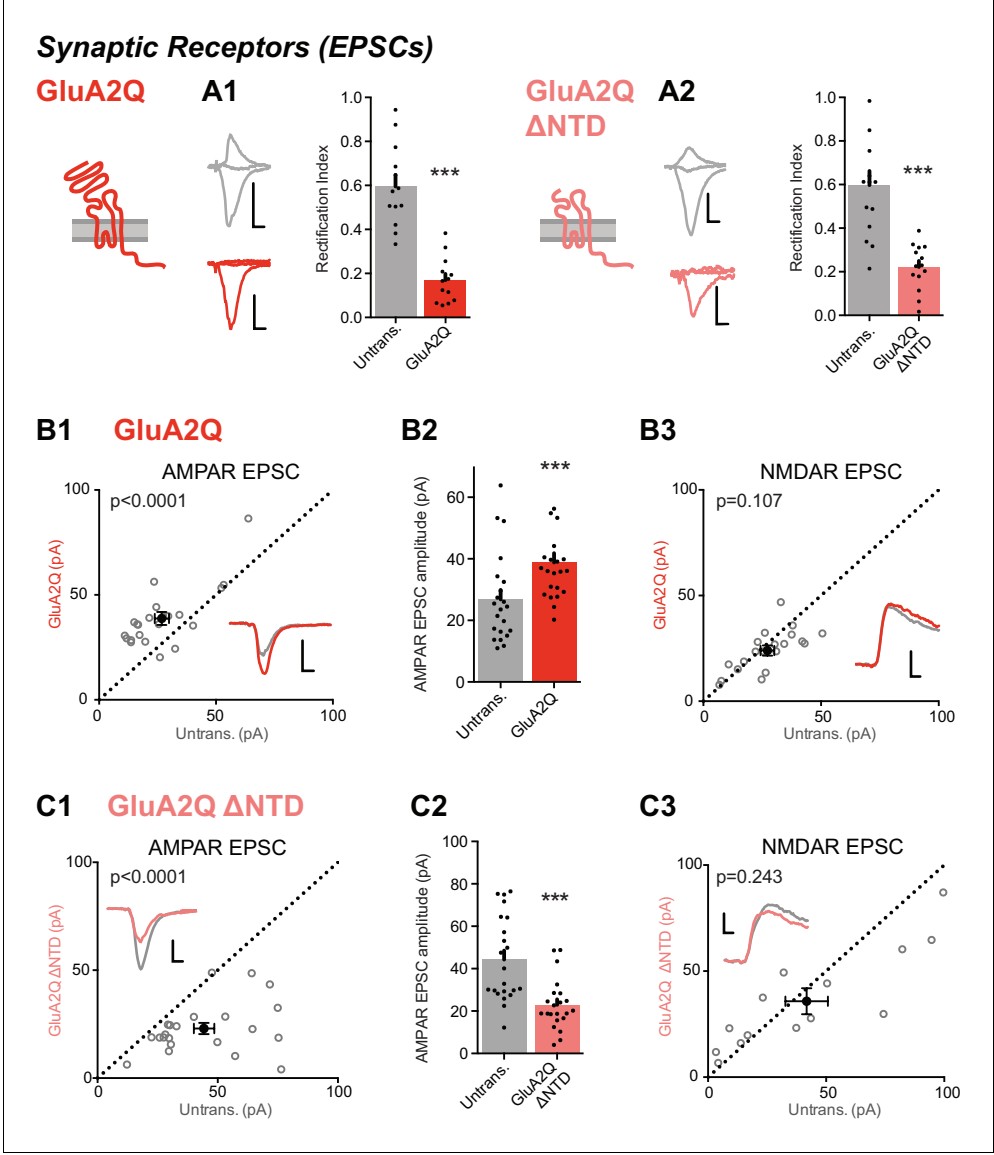

**Figure 2.** Expression of NTD deleted GluA2 causes large reduction in synaptic currents. (**A**) Synaptic RI measured from pairs of untransfected and transfected cells indicate both homomeric GluA2Q and GluA2Q ΔNTD are inserted into synapses. (**A1**) Untrans., 0.60 ± 0.19; GluA2Q, 0.17 ± 0.10; n = 13 pairs; paired t-test, p=0.001. (**A2**) Untrans., 0.60 ± 0.25; GluA2Q ΔNTD, 0.22 ± 0.11; n = 16 pairs; paired t-test, p<0.0001. Sample traces and construct schematics are shown on the left. Scale bar = 10 ms, 50 pA (grey and GluA2Q) or 20 pA (GluA2Q ΔNTD). (**B–C**) Scatter plots and bar charts of EPSC amplitudes from pairs of cells, showing single pairs (open circles) and mean values ± SEM (filled circles). Sample traces inset. Scale bar = 10 ms, 30 pA. (**B1**) GluA2Q-expressing cells have increased AMPAR EPSCs relative to untransfected cells (untrans.: 26.8 ± 3.1 pA; GluA2Q: 38.8 ± 3.0 pA; n = 22 pairs; paired t-test, p<0.0001). (**B2**) Bar chart of AMPAR EPSCs from **B1**. (**B3**) NMDAR-mediated EPSCs remain unchanged (untrans.: 27.2 ± 2.7 pA; GluA2Q: 24.0 ± 2.2 pA; n = 20; paired t-test, p=0.107). (**C1**) GluA2Q ΔNTD-expressing cells show AMPAR EPSC amplitude depression (untrans.: 44.3 ± 4.3 pA; GluA2Q ΔNTD: 23.0 ± 2.5 pA; n = 22; paired t-test, p<0.0001). (**C2**) Bar chart of AMPAR EPSCs from **C1**. (**C3**) NMDAR EPSCs show no amplitude change (untrans.: 41.8 ± 9.0 pA; GluA2Q ΔNTD: 35.8 ± 6.1 pA; n = 14; paired t-test, p=0.243).

The following figure supplement is available for figure 2:

**Figure supplement 1.** Measurement of paired-pulse ratio and spine density.

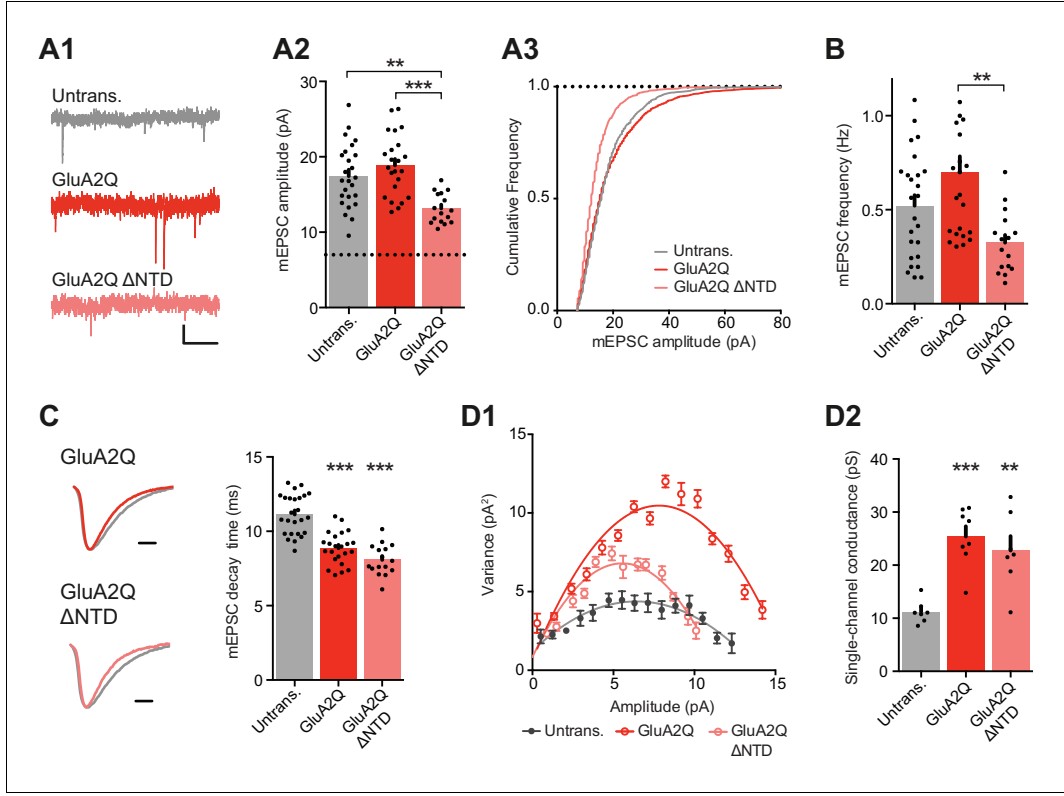

**Figure 3.** NTD deleted GluA2 is detrimental to spontaneous transmission. (**A1**) Example traces of mEPSCs recorded from untransfected, GluA2Q and GluA2Q ΔNTD-expressing cells. Scale bar = 0.5 s, 5 pA. (**A2**) Bar chart of mEPSC amplitude with event detection limit indicated (dotted line) (untransfected: 17.5 ± 0.8 pA (n = 25 cells); GluA2Q: 18.9 ± 0.8 pA (n = 23); GluA2Q ΔNTD: 13.2 ± 0.5 pA (n = 16); One-way ANOVA, p<0.0001). (**A3**) Cumulative frequency distribution of mEPSC amplitude data from **A2**. (**B**) Bar chart of mEPSC frequency (untransfected: 0.52 ± 0.06 Hz; GluA2Q: 0.70 ± 0.08 Hz; GluA2Q ΔNTD: 0.32 ± 0.04 Hz; One-way ANOVA, p=0.002). (**C**) Example traces of scaled mEPSCs from untransfected (grey) and GluA2 construct-expressing cells. Scale bar = 3 ms. Bar chart shows cell averaged mEPSC decay times (untrans.: 11.14 ± 0.27 ms (n = 25); GluA2Q: 8.86 ± 0.23 ms (n = 23); GluA2Q ΔNTD: 8.08 ± 0.26 ms (n = 16); One-way ANOVA, p<0.0001). (**D1**) Amplitude vs. variance plot for non-stationary fluctuation analysis (NSFA) of scaled mEPSCs with parabolic fits from representative cells. (**D2**) Single-channel conductance of synaptic AMPARs show comparable conductance between GluA2Q and GluA2Q ΔNTD-expressing cells (untransfected: 11.2 ± 0.82 pS (n = 7); GluA2Q: 25.4 ± 1.92 pS (n = 8); GluA2Q ΔNTD: 22.8 ± 2.70 pS (n = 7); One-way ANOVA, p=0.0002).

## Receptor mobility is regulated via the GluA2 NTD

To test whether the NTD stabilizes AMPARs at synapses, we assayed receptor mobility using fluorescence recovery after photobleaching (FRAP) in cultures of dissociated hippocampal neurons. GluA2Q and GluA2Q ΔNTD were tagged at their N-termini with Super Ecliptic pHluorin (SEP), a pH-sensitive GFP variant. SEP is quenched at low-pH (as found in endosomal transport vesicles) facilitating visualization of surface-expressed receptors, which are exposed to neutral pH (*Figure 4A*) (*Ashby et al., 2006*; *Makino and Malinow, 2009*).

In dendritic spines, SEP-GluA2 recovered from bleaching with a time constant of $\tau_{rec}$=197 s, and recovery was incomplete after 10 min with an immobile fraction of ~40% (at t = 600 s; *Figure 4B*). These values are in line with previous studies (*Kerr and Blanpied, 2012*; *Makino and Malinow, 2009*; *Zhang et al., 2013*). In sharp contrast, recovery of SEP-GluA2 ΔNTD was rapid ($\tau_{rec}$= 65 s) and the immobile fraction at 600 s was reduced to just 7%. Thus, NTD-deleted receptors are poorly confined in spines when compared to full-length GluA2Q. Interestingly, this difference was specific to spine fluorescence: in extra-synaptic (dendritic) regions, SEP-GluA2 exhibited rapid ($\tau_{rec}$ = 98 s) and almost complete recovery (immobile fraction = 18%), which was not significantly different to

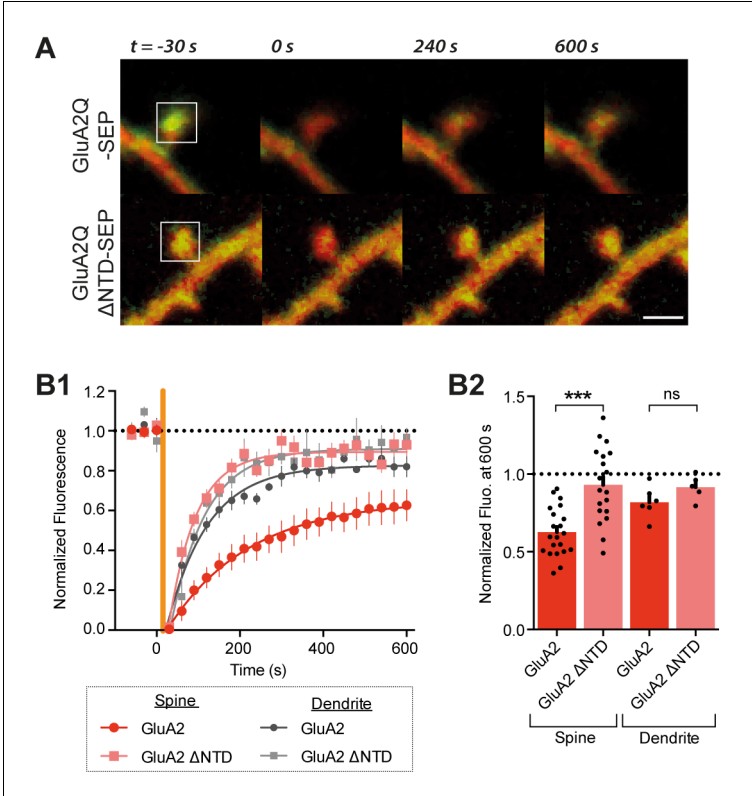

**Figure 4.** The GluA2 NTD controls synaptic immobilisation. (**A**) Example images of FRAP on dendritic spine from cells expressing SEP-tagged AMPAR constructs, where t = 0 indicates time and square indicates location of photobleaching. Red channel: cytosolic mCherry; green: SEP fluorescence. Scale bar = 1 μm **B1**, SEP fluorescence over time in bleached regions of dendrite or spine, normalized to pre-bleaching fluorescence. Orange vertical line indicates onset of photobleaching (time constant of fit τ: spine GluA2 = 197.3, spine GluA2 ΔNTD = 65.4, dendritic GluA2 = 98.1, dendritic GluA2 ΔNTD = 83.1). (**B2**), Fluorescence at 600s averaged by cell shows greater recovery for GluA2Q ΔNTD than full-length GluA2Q (spine GluA2Q: 0.63 ± 0.03 (n = 22 cells); spine GluA2Q ΔNTD: 0.93 ± 0.05 (n = 19); dendrite GluA2Q: 0.80 ± 0.04 (n = 6); dendritic GluA2Q ΔNTD: 0.92 ± 0.04 (n = 5); One-way ANOVA, p<0.0001).

that of SEP-GluA2 ΔNTD (**Figure 4B**). Moreover, diffusion in dendrites resembled the behavior of spine localized SEP-GluA2 ΔNTD. These data support the hypothesis that the NTD plays a role in specifically stabilizing AMPARs at postsynaptic sites.

## GluA1 requires the NTD for synaptic delivery

We next extended our experiments to GluA1, which differs from GluA2 in primary sequence chiefly in the NTD and the CTD. Each subunit exhibits different trafficking properties that have been exclusively ascribed to their CTD (**Shepherd and Huganir, 2007**; **Shi et al., 2001**).

Similar to GluA2, NTD deletion (**Figure 1—figure supplement 1B3**) does not affect GluA1 trafficking to the cell surface, and the RI of the NTD-deleted receptors was comparable to GluA1 in somatic patches of CA1 pyramidal cells (RI GluA1: 0.23 ± 0.03, GluA1 ΔNTD: 0.19 ± 0.02) (**Figure 5A**). Moreover, as was the case for GluA2, the KA/Glu ratio of neurons expressing either GluA1 construct was similar to untransfected cells (**Figure 5B1**) and current amplitudes were approximately doubled on expression of GluA1 or GluA1 ΔNTD (**Figure 5B2**). Again, TARPs do not appear to be limiting and expression levels of exogenous subunits were comparable.

Contrasting with previous studies (**Hayashi et al., 2000**; **Shi et al., 2001**), we find that the RI of GluA1 expressing cells is consistently lower than untransfected cells, (RI - Untrans.: 0.57 ± 0.04, GluA1: 0.33 ± 0.03), although not to the same extent as GluA2 (**Figure 5C1** versus **Figure 2A1**).

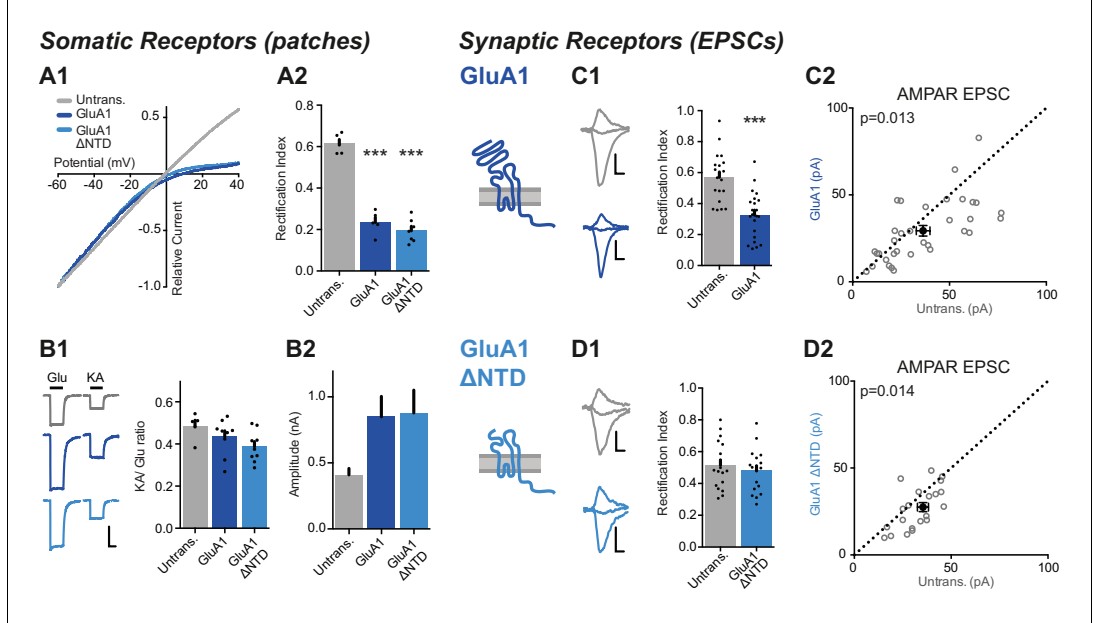

**Figure 5.** The NTD is essential for synaptic anchoring of GluA1. (**A1**) I/V relationships of glutamate-evoked AMPAR-mediated current from outside-out patches of untransfected, GluA1 and GluA1 ΔNTD expressing cells. (**A2**) Average RI of surface currents from above neurons (untrans.: 0.61 ± 0.05 (n = 5); GluA1: 0.23 ± 0.03 (n = 5); GluA1 ΔNTD: 0.19 ± 0.02 (n = 7); One-way ANOVA, p<0.0001). (**B1**) KA/Glu ratio from somatic patches is unchanged on GluA1 construct overexpression (KA/Glu: untrans.: 0.47 ± 0.02 (n = 5); GluA1: 0.43 ± 0.03 (n = 9); GluA1 ΔNTD: 0.39 ± 0.02 (n = 9); One-way ANOVA, p=0.16). Example traces showing glutamate (Glu) and kainic acid (KA) application are shown left. Scale bar = 50 ms and 300 pA. (**B2**) AMPAR surface patch amplitudes are similarly elevated on GluA1 or GluA1 ΔNTD overexpression (untrans.: 421 ± 90 pA; GluA1: 849 ± 154 pA; GluA1 ΔNTD: 878 ± 173 pA). (**C1**) Synaptic RI from pairs of untransfected and GluA1-expressing cells (untransfected: 0.57 ± 0.04; GluA1: 0.33 ± 0.03; n = 20; paired t-test, p=0.0001), with example traces and construct schematic shown on the left. Scale bars for panels **C** and **D** = 10 ms and 15 pA. C2 Scatter plot of AMPAR EPSC amplitudes from pairs of untransfected and GluA1-expressing cells (untransfected: 36.4 ± 3.5 pA; GluA1: 29.4 ± 3.0 pA; n = 34; paired t-test, p=0.013). (**D1**) Synaptic RI from pairs of untransfected and GluA1 ΔNTD-expressing cells (untransfected: 0.51 ± 0.04; GluA1 ΔNTD: 0.48 ± 0.03; n = 18; paired t-test, p=0.072), with example traces shown on the left. (**D2**) Scatter plot of AMPAR EPSC amplitudes from pairs of untransfected and GluA1 ΔNTD-expressing cells (untransfected: 32.6 ± 2.1 pA; GluA1 ΔNTD: 26.9 ± 2.7 pA; n = 21; paired t-test, p=0.014).

Strikingly, in contrast to GluA2 ΔNTD, GluA1 ΔNTD expression was unable to change the rectification index, which remained comparable to untransfected cells (*Figure 5D1*), suggesting that synaptic anchoring of GluA1 was completely dependent upon its NTD. Unlike GluA2Q, EPSC amplitudes were not elevated upon GluA1 expression but were slightly decreased relative to untransfected neurons; an effect that was evident for both GluA1 constructs (*Figure 5C2 and D2*). These data demonstrate an essential role for the GluA1 NTD in synaptic incorporation.

## NTD dependent anchoring maintains synaptic AMPARs in a knockout background

To corroborate these observations, we also examined NTD-dependent anchoring of GluA1 and GluA2 in an AMPAR null background, using organotypic slices from conditional GluA1-3 knockout mice (*Gria1^{lox/lox}; Gria2^{lox/lox}; Gria3^{lox/lox}*, denoted *Gria1-3fl*) (*Lu et al., 2009*). Both overexpression and knockout/rescue approaches provide complementary information on receptor function. While knockout/rescue permits unequivocal quantification and interpretation of receptor contributions without interference from endogenous receptors, overexpression prevents any compensatory effects of receptor removal. Additionally, competition with endogenous subunits for synaptic slots facilitates identification of subtle deficits caused by receptor mutation that would be of lesser consequence using a knockout approach.

*Gria1-3* genes were excised by viral injection of Cre-recombinase into P0 mouse pups (AAV-Cre-GFP) and AMPAR null neurons were rescued by single-cell electroporation of AMPAR constructs into organotypic slices 12 days later (*Figure 6—figure supplement 1A*). Successful GluA1-3 deletion

was confirmed, as CA1 neurons expressing Cre-GFP alone showed almost complete loss of AMPAR responses, as assayed using the ratio of AMPAR and NMDAR EPSC amplitudes (AMPAR/NMDAR) (*Figure 6—figure supplement 1B*). GluA2Q transfection rescued the AMPAR EPSC to levels comparable with uninfected (Cre-negative) neurons of each paired recording (*Figure 6A1*), whereas transfection of GluA2Q ΔNTD did not (*Figure 6A2*). Normalization to the untransfected cell of each pair revealed that GluA2Q ΔNTD rescue was less than half that of GluA2Q (Relative AMPAR/NMDAR - GluA2Q: 0.99 ± 0.10, GluA2Q ΔNTD: 0.44 ± 0.05; *Figure 6A3*). This difference between GluA2Q and GluA2Q ΔNTD responses closely matches our previous observations (*Figure 2B,C*).

Contrasting with GluA2Q, a complete rescue could not be achieved with GluA1, and rescue was further impaired when expressing the GluA1 ΔNTD mutant (*Figure 6B1–3*), as seen with the

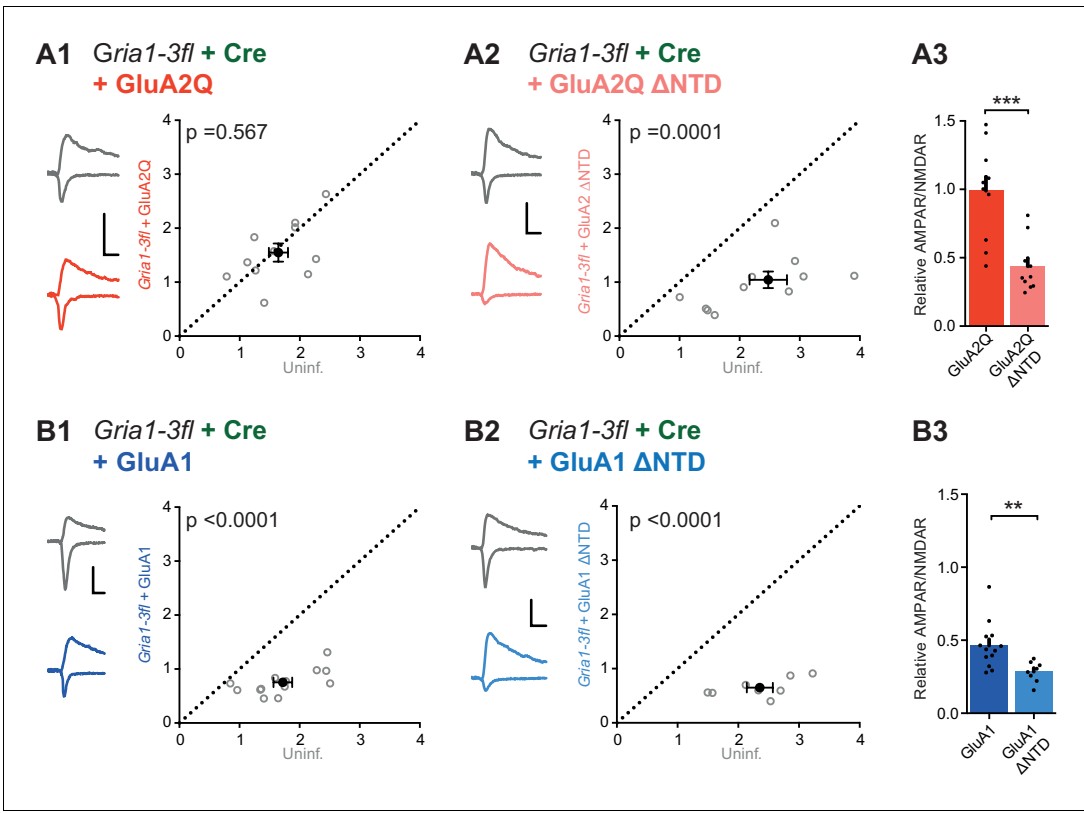

**Figure 6.** NTD dependent interactions enhance synaptic AMPAR anchoring in a knockout background. Paired recordings from *Gria1-3fl* neurons infected with AAV-Cre and rescued with AMPAR subunits, or uninfected and untransfected (uninf.). Example traces show current responses at −60 mV (AMPAR) and +40 mV (NMDAR) holding potentials. Scale bars = 30 ms and 50 pA. (**A1**) Rescue with GluA2Q restores the ratio of AMPAR to NMDAR currents to levels of uninfected neurons (AMPAR/NMDAR, (n = 11 pairs): uninf.: 1.64 ± 0.16; GluA2Q: 1.55 ± 0.17; paired t-test, p=0.567). (**A2**) Rescue with GluA2Q ΔNTD cannot fully restore AMPAR currents relative to NMDAR (AMPAR/NMDAR, (n = 11 pairs): uninf.: 2.28 ± 0.26; GluA2Q ΔNTD 0.97 ± 0.15; paired t-test, p=0.0001). (**A3**) Normalization of synaptic currents to uninfected cells reveals that GluA2Q NTD deletion reduces synaptic AMPAR rescue (Relative AMPAR/NMDAR ratio: GluA2Q: 0.99 ± 0.10; GluA2Q ΔNTD: 0.44 ± 0.05; unpaired t-test, p=0.0001). (**B1**) Rescue of synaptic currents by GluA1 transfection (AMPAR/NMDAR, (n = 13 pairs): uninf.: 1.72 ± 0.15; GluA1: 0.75 ± 0.06; paired t-test, p<0.0001). (**B2**) Rescue of synaptic currents with GluA1 ΔNTD (AMPAR/NMDAR, (n = 9 pairs): uninf.: 2.48 ± 0.23; GluA1 ΔNTD: 0.83 ± 0.19; paired t-test, p<0.0001). (**B3**) GluA1 rescues synaptic currents to a greater extent than GluA1 ΔNTD (Relative AMPAR/NMDAR ratio: GluA1: 0.46 ± 0.04; GluA1 ΔNTD: 0.28 ± 0.03; unpaired t-test, p=0.007).

The following figure supplement is available for figure 6:

**Figure supplement 1.** Overview and characterization of conditional AMPAR knockout in *Gria1-3fl* organotypic slices.

overexpression data (*Figure 5C and D*). These results underscore the AMPAR's dependence on its NTD for synaptic anchoring.

## The GluA2 NTD enhances GluA1 delivery into synapses

To examine the subunit selectivity of the NTD further, we swapped this domain between GluA1 and GluA2 and expressed the resulting swap mutants in WT neurons. Both mutants readily trafficked to the cell surface and altered the RI in somatic patches similar to the WT subunits (*Figure 7—figure supplement 1A*). Transplanting the GluA2 NTD onto GluA1 (GluA1 +A2NTD) enhanced synaptic inward rectification (RI 0.22 ± 0.02, *Figure 7A1*) and increased response amplitudes relative to GluA1 WT (*Figure 7A2 and A3*), further demonstrating that the GluA2 NTD is able to promote AMPAR incorporation into synapses. In line with this, in neurons expressing GluA2Q +A1NTD (i.e. the GluA2 NTD swapped for that of GluA1) response amplitudes were reduced relative to WT GluA2Q, approaching values obtained with GluA1 (*Figure 7B2 and B3*). Therefore, the NTD of

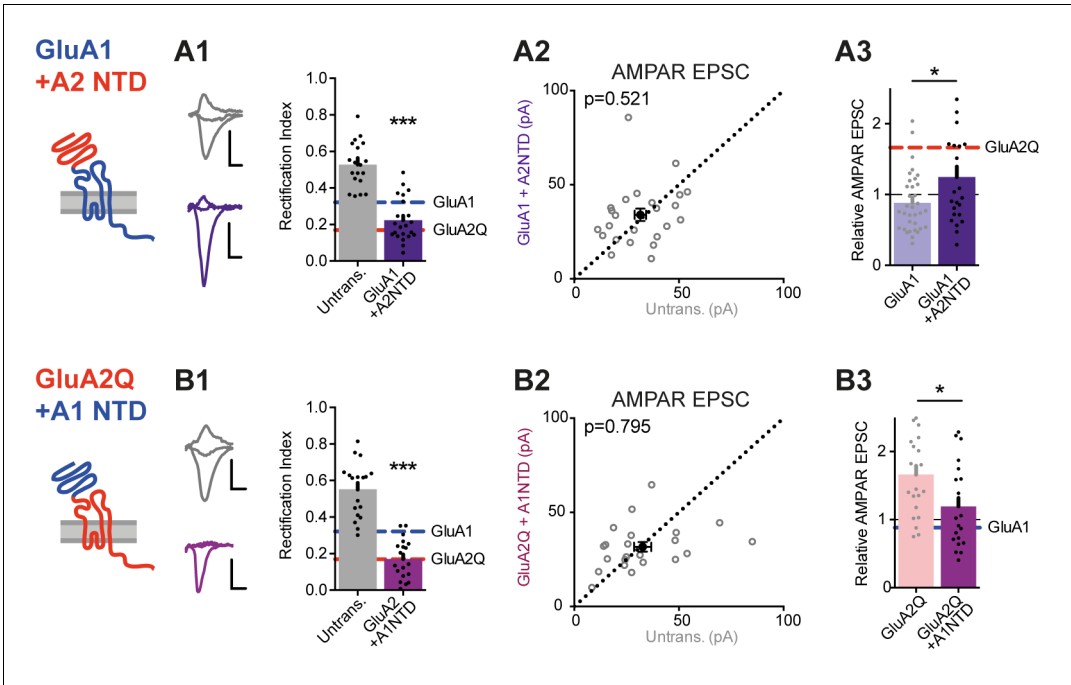

**Figure 7.** NTD-dependent synaptic anchoring is subunit-specific. Synaptic EPSC properties of neurons expressing chimeric AMPAR constructs formed by exchanging NTD sequences (see construct schematics). (**A1**) Synaptic RI from pairs of untransfected cells and cells expressing GluA1 +A2NTD (untransfected: 0.56 ± 0.04; GluA1 +A2NTD: 0.22 ± 0.02; n = 23; paired t-test, p<0.0001), with example traces and construct schematic shown on the left. GluA1 and GluA2Q RI values are indicated for reference. Scale bars = 10 ms and 20 pA. (**A2**) Scatter plot of AMPAR EPSCs from pairs of untransfected and transfected cells expressing GluA1 +A2NTD (untrans.: 31.6 ± 2.7 pA; GluA1 +A2NTD: 34.0 ± 3.3 pA; n = 24; paired t-test, p=0.521). (**A3**) Bar chart of AMPAR EPSC amplitudes of transfected cells normalized to untransfected cells of paired recordings (GluA1: 0.88 ± 0.07 (n = 34); GluA1 +A2NTD: 1.25 ± 0.15 (n = 24); unpaired t-test, p=0.017). Value for GluA2Q is indicated by red line for reference. (**B1**) Synaptic RI from pairs of untransfected and GluA2Q +A1NTD-expressing cells (untransfected: 0.55 ± 0.03; GluA2Q +A1NTD: 0.17 ± 0.03; n = 19; paired t-test, p<0.0001), with example traces shown on the left. (**B2**) Scatter plot of AMPAR EPSCs from pairs of untransfected and GluA2Q +A1NTD (untrans.: 32.7 ± 4.1 pA; GluA2Q +A1NTD: 31.6 ± 2.6 pA; n = 22; paired t-test, p=0.795). (**B3**) Bar chart of AMPAR EPSCs amplitudes normalized to untransfected cell of paired recording (GluA2Q: 1.66 ± 0.13 (n = 22); GluA2Q +A1NTD: 1.19 ± 0.13 (n = 22); unpaired t-test, p=0.013). Value for GluA1 is indicated by blue line for reference.

The following figure supplement is available for figure 7:

**Figure supplement 1.** Investigation of GluA2Q ΔNTD + A1CTD.

GluA2 appears to confer a unique 'synapto-sticky' phenotype that efficiently drives the receptor into synapses.

We note that the rectification index of GluA2Q +A1NTD expressing cells was comparable to GluA2Q WT (*Figure 7B1*). This is unsurprising, as GluA2 can traffic to synapses without any NTD (see GluA2Q *ΔNTD*; *Figure 2A2*). NTD-independent GluA2 trafficking, causing the strong rectification and synaptic depression of GluA2Q *ΔNTD*-expressing cells is likely to be caused in part by its C-terminal tail. Indeed, replacing the CTD of GluA2Q *ΔNTD* with that of GluA1 (GluA2Q-*ΔNTD* +A1CTD) greatly reduced inward rectification and alleviated synaptic depression of postsynaptic responses (*Figure 7—figure supplement 1B*).

## LTP expression requires the NTD

LTP expression requires recruitment of additional AMPARs to synapses (*Huganir and Nicoll, 2013*; *Kessels and Malinow, 2009*). This has been long associated with the GluA1 subunit (*Hayashi et al., 2000*; *Zamanillo et al., 1999*), and explained by activity-dependent trafficking requiring the GluA1 CTD (*Shi et al., 2001*). However, this model has recently been challenged (*Granger et al., 2013*, see also *Kim et al., 2005*).

Our experiments show that the GluA1 NTD is essential for synaptic anchoring under basal conditions (*Figure 5*). To assess whether the NTD also plays a role in synaptic plasticity, we utilized two potentiation protocols to compare GluA1 and GluA1 *ΔNTD*: (i) expression of constitutively active CaMKII (tCaMKII) (*Hayashi et al., 2000*), a kinase essential for LTP (*Hell, 2014*), and (ii) electrical stimulation using a pairing protocol.

Neurons transfected with tCaMKII gave rise to significantly enhanced EPSCs (*Figure 8—figure supplement 1A1*), in line with earlier work (*Hayashi et al., 2000*). Expression of tCaMKII together with GluA1 similarly potentiated responses (*Figure 8A1*) and showed inward rectification (*Figure 8A2* cf. *Figure 8—figure supplement 1A2*). RI does not appear to differ from expression of GluA1 alone (*Figure 5C1*), indicating that the potentiation is mediated by both recombinant GluA1 homomers, and native AMPARs (such as GluA1/2 heteromers). However, synapses expressing tCaMKII with GluA1 *ΔNTD* failed to potentiate (*Figure 8B1*). Given that the EPSCs showed rectification (*Figure 8B2*), the potentiating stimulus appears to drive these receptors into the synapse, as described previously (*Hayashi et al., 2000*), yet without their NTD they are unable to maintain a potentiated state.

To investigate this observation further, we examined LTP in neurons expressing GluA1 either with or without its NTD. Using a pairing protocol, LTP could be reliably induced in both untransfected cells (*Figure 8—figure supplement 1B1*) and cells expressing GluA1 (*Figure 8C*), with enhanced transmission maintained for at least 45 min after induction. However, although neurons in which the extrasynaptic pool of receptors contained GluA1 *ΔNTD* showed a transient potentiation, EPSC amplitudes had returned to baseline levels 30–35 min after induction (*Figure 8C*). In a subset of recordings, a second stimulation pathway was included, in which no potentiation was induced (*Figure 8—figure supplement 1B2*). This pathway showed similar amplitudes and no LTP induction in either condition, confirming the stability of recordings, yet the effect on LTP seen in test pathways was clearly exhibited. Thus, taken together with tCaMKII expression data, NTD-dependent interactions are essential for AMPA receptor anchoring, which is a prerequisite for synaptic potentiation.

## Discussion

AMPAR insertion into synapses has emerged as a central mechanism underlying the expression of LTP (*Durand et al., 1996*; *Isaac et al., 1995*; *Liao et al., 1995*). Regulation of AMPAR trafficking to (and from) synapses involves lateral diffusion (*Choquet and Triller, 2013*) and vesicular trafficking (*Newpher and Ehlers, 2008*). These events have been mostly ascribed to the receptor CTD: 50–80 residue long cytosolic extensions that vary in sequence, are selectively phosphorylated, and interact with scaffolding and actin-binding proteins in a subunit-selective manner (*Anggono and Huganir, 2012*; *Shepherd and Huganir, 2007*). Recent work suggests that the tails only play a modulatory role in LTP, raising the possibility that other segments of the receptor are essential for synaptic targeting (*Granger et al., 2013*). Here we demonstrate that the N-terminal domain is a central player in this process, and participates in AMPAR anchoring at synapses in a subunit-selective manner.

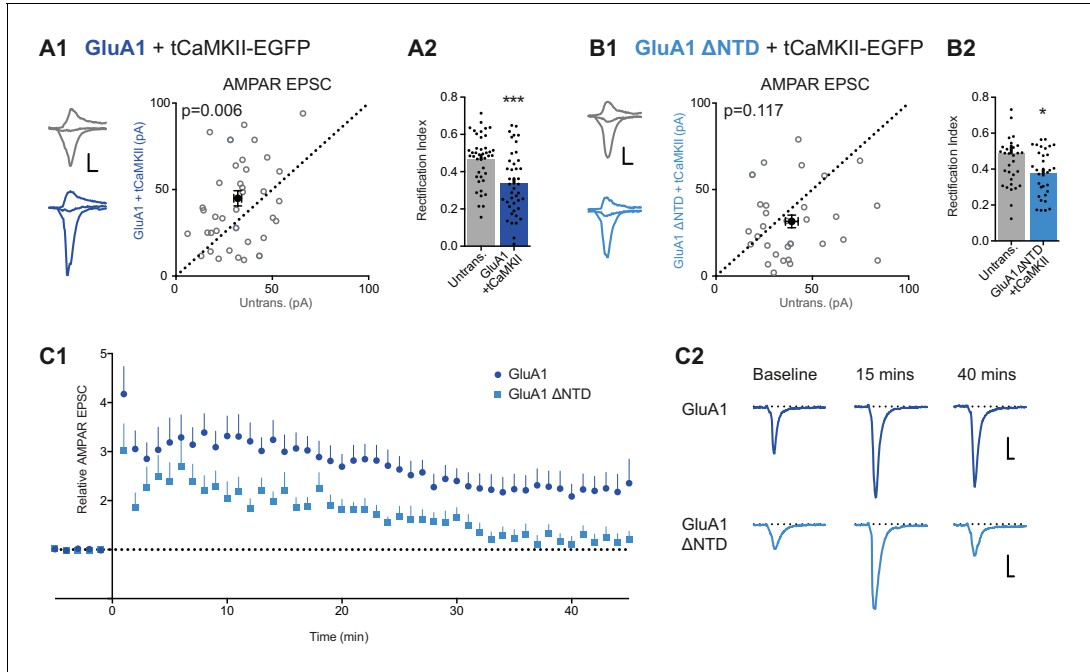

**Figure 8.** LTP is impaired without the NTD of GluA1. (**A1**) Co-expression of GluA1 and tCaMKII increases AMPAR EPSC amplitude (untransfected: 32.1 ± 2.0 pA; GluA1+tCaMKII: 44.9 ± 4.4 pA; n = 42; paired t-test, p=0.006). Scale bar = 10 ms and 20 pA. (**A2**) RI from above cells (untransfected: 0.47 ± 0.02; GluA1 + tCaMKII: 0.34 ± 0.03; n = 42; paired t-test, p=0.0003). (**B1**) Potentiation mediated by tCaMKII is impaired in GluA1 *ΔNTD* expressing cells (untransfected: 39.3 ± 3.3 pA; GluA1 *ΔNTD* + tCaMKII: 31.6 ± 3.7 pA; n = 32; paired t-test, p=0.117). Scale bar = 10 ms and 20 pA. (**B2**) RI from above cells (untransfected: 0.49 ± 0.04; GluA1 *ΔNTD* + tCaMKII: 0.38 ± 0.02; n = 33; paired t-test, p=0.019). (**C1**) AMPAR EPSC amplitudes from cells expressing GluA1 or GluA1 *ΔNTD* over time, averaged in one-minute bins. At time = 0 LTP was induced using a pairing protocol (2 Hz, 100 s at −10 mV holding potential). EPSC amplitudes are normalized to pre-induction amplitude. LTP is maintained past 35 min in cells expressing GluA1, but not GluA1 *ΔNTD*. Normalized amplitude at 45 mins: GluA1: 2.35 ± 0.49 (n = 16); GluA1 *ΔNTD*: 1.20 ± 0.17 (n = 14). (**C2**) Representative AMPAR EPSC traces from cells expressing GluA1 or GluA1 *ΔNTD*. Traces show EPSCs before induction (baseline) and 15 and 40 min after induction. Scale bar = 10 ms and 20 pA.

The following figure supplements are available for figure 8:

**Figure supplement 1.** Synaptic potentiation using tCaMKII and an electrical pairing protocol.

**Figure supplement 2.** The sequence diversity of the AMPAR NTD mediates subunit-selective synaptic anchoring.

AMPARs rapidly diffuse in the plane of the membrane and are trapped at postsynaptic sites upon LTP (*Choquet and Triller, 2013*; *Opazo et al., 2012*). Receptor trapping and clustering may occur selectively opposite presynaptic release sites, to ensure optimal receptor activation on neurotransmitter release (*Lisman et al., 2007*; *Raghavachari and Lisman, 2004*; *Tang et al., 2016*). The NTD, which projects about mid-way into the synaptic cleft, is ideally suited to engage interaction partners. Structural data (*Dürr et al., 2014*; *Herguedas et al., 2016*; *Meyerson et al., 2014*; *Nakagawa et al., 2005*; *Sukumaran et al., 2011*) and receptor simulations (*Dutta et al., 2015*; *Krieger et al., 2015*) have shown that this domain layer is highly dynamic and could therefore 'sample' the local environment for binding partners (*García-Nafría et al., 2016a*). Some NTD-interactors have been identified: secreted pentraxins (*O'Brien et al., 1999*; *Sia et al., 2007*), the adhesion molecule N-cadherin (*Saglietti et al., 2007*), and AMPAR auxiliary subunits (*Cais et al., 2014*), but whether any of these molecules participate in selective synaptic anchoring remains to be established.

The NTD is highly sequence diverse (*Figure 8—figure supplement 2A*) with only 56% sequence identity between subunits in rodents. Our data are most compatible with an NTD-anchoring mechanism that is subunit-selective, mediated by this diversity, and supports some of the previous work on subunit-specific AMPAR trafficking (*Malinow et al., 2000*; *Shi et al., 2001*). In line with these

studies, we show that GluA2 integrates into synapses more efficiently than GluA1, an observation that is supported by previous AMPAR knockout data. In a conditional GluA2/3 knockout, GluA1 homomers are unable to maintain full synaptic transmission, despite providing a complete extrasynaptic pool of receptors (*Lu et al., 2009*), a deficit that is alleviated with GluA2 also present. As we show that replacement of the GluA1 NTD with that of GluA2 facilitates more robust incorporation into the synapse, N-terminal domain interactions appear to effectuate this subunit-specific anchoring. We also demonstrate that the GluA2 CTD facilitates synapse targeting, but alone, it is unable to stably position the receptor in the absence of the NTD. The depression in basal transmission seen in GluA2Q ΔNTD expressing cells presumably occurs through sequestration of critical, subunit-specific interactors from native receptors, as this can be alleviated through CTD exchange with GluA1 (*Figure 7—figure supplement 1B*). Interestingly, the GluA1 CTD is insufficient to deliver GluA1 to synapses and this subunit strictly depends on its NTD for targeting and anchoring.

In potentiating the synapse the GluA1 CTD has been described as both essential (*Hayashi et al., 2000*; *Shi et al., 2001*) and dispensable (*Granger et al., 2013*). While we observe that tCaMKII expression appears to drive synaptic incorporation of GluA1 ΔNTD, without the AMPAR NTD, potentiation does not occur. Interestingly, potentiation cannot be achieved by native receptors in these cells, indicating a possible sequestration of important CTD interactors by GluA1 ΔNTD in a similar manner to GluA2 under basal conditions, which is reminiscent of the conclusions of *Shi et al. (2001)*. It is clear from somatic patch recordings that exogenously expressed receptors comprise the vast majority of the extrasynaptic pool. When LTP is induced in GluA1 ΔNTD-expressing cells, rapid short-term potentiation is observed, yet this potentiation cannot be maintained. The transient short-term potentiation most likely requires NTD-independent AMPAR trafficking mechanisms, such as TARP phosphorylation (*Opazo et al., 2010*), highlighting the fine interplay of interactions that dictates AMPAR delivery to synapses. However, the synaptic rearrangements required for LTP appear critically dependent on the AMPAR NTD. We propose that a key requirement for LTP expression is stable receptor anchorage via the NTD, when the synapse is rearranged to a potentiated state.

Based on these data, we hypothesize that CTD interactions are important in accruing receptors at postsynaptic sites, but NTD interactions are key for positioning or stabilizing the AMPAR for effective transmission (*Figure 8—figure supplement 2B*). The GluA2 NTD's affinity for the synaptic sites may allow critical control of synaptic signaling. As GluA2 renders AMPARs calcium-impermeable, this interaction has the potential to bias for GluA2-containing receptors, preventing excitotoxicity and controlling the potential signaling of calcium-permeable AMPARs (*Cull-Candy et al., 2006*).

Our results shed light on some controversies in the literature. Whereas GFP-GluA1 is unable to traffic to the synapse without a potentiating stimulus (*Hayashi et al., 2000*; *Shi et al., 2001*), constitutive synaptic trafficking of GluA1 has been described, with the discrepancy being attributed to the N-terminal GFP tag (*Granger et al., 2013*, but see *Nabavi et al., 2014*). This can be explained by the essential requirement for the GluA1 NTD that we describe (*Figure 5*). Similarly, we report an increase in AMPAR EPSC amplitude upon expression of GluA2Q (*Figure 2B*), likely mediated by the increased channel conductance, which was not seen using N-terminally GFP-tagged GluA2Q (*Shi et al., 2001*). The authors reported changes in rectification (and hence synaptic delivery of GluA2-GFP) but their construct did not give rise to elevated AMPAR amplitudes. We tested whether the presence of the tag could explain the discrepancy with our data. Indeed, inserting GFP upstream of the GluA2 NTD reduced current amplitudes to levels of untransfected control neurons (*Figure 8—figure supplement 1C*), supporting the hypothesis that the GFP tag interferes with GluA2 synaptic anchoring. This effect has implications for interpretation of our FRAP data, but given the clear functional difference between GFP-GluA2 and GluA2 ΔNTD, it appears that the GFP tag impairs but does not abolish GluA2 anchorage.

Recent studies have highlighted a fundamental role for glutamate receptor NTDs in synapse operation and architecture (*Elegheert et al., 2016*; *Matsuda et al., 2016*). In each case, presynaptic cell adhesion molecules have been identified as critical interaction partners. Given the recently emerging role for synaptic adhesion molecules in LTP (*Aoto et al., 2013*; *Shipman and Nicoll, 2012*; *Soler-Llavina et al., 2013*) and structured alignment of the postsynapse with presynaptic release machinery (*Tang et al., 2016*), transsynaptic interactions are likely to play a key role in controlling AMPAR signaling. The N-terminal domain now emerges as a prime candidate to mediate these effects.

# Materials and methods

## Constructs

Rat sequence AMPAR subunits GluA1 and GluA2 (flip and R/G edited) were expressed from the pRK5 vector. All cloning procedures were performed using IVA cloning (*García-Nafría et al., 2016b*). GluA2 mutation R586Q was used for all experiments (GluA2Q). GluA1 and GluA2 ΔNTD were created by simultaneous deletion of the NTD coding region (GluA1 residues 1–373, GluA2 1–377) and replacement with a c-myc epitope sequence immediately after the signal sequence. The tCaMKII-EGFP construct was a gift from José Esteban, and has been described previously (*Shi et al., 2001*). GluA1/2 NTD swap constructs were created by exchange of NTD and NTD-LBD linker sequences (GluA1 residues 1–390, GluA2 1–394) GluA2 ΔNTD +A1CTD tail swap construct was created by exchange of the entire C-terminal sequence of GluA2 (813–862) with that of GluA1 (809–889). Super-ecliptic pHluorin (SEP) or GFP tagged GluA2 was produced by insertion of fluorescent protein coding region between the third and fourth residues of the mature GluA2 protein, preceded and followed by an 'Ala-Ser' dipeptide linker. SEP sequence was a gift from Jonathan Hanley.

## Organotypic slice cultures

All procedures were carried out under PPL 70/8135 in accordance with UK Home Office regulations. Experiments conducted in the UK are licensed under the UK Animals (Scientific Procedures) Act of 1986 following local ethical approval.

Organotypic slice cultures were prepared as described previously (*Stoppini et al., 1991*). Briefly, hippocampi from P6-8 C57/Bl6 mice were isolated in high-sucrose Gey's balanced salt solution containing (in mM): 175 Sucrose, 50 NaCl, 2.5 KCl, 0.85 $NaH_2PO_4$, 0.66 $KH_2PO_4$, 2.7 $NaHCO_3$, 0.28 $MgSO_4$, 2 $MgCl_2$, 0.5 $CaCl_2$ and 25 glucose at pH 7.3. Hippocampi were cut into 300 μm thick slices using a McIlwain tissue chopper and cultured on Millicell cell culture inserts (Millipore Ltd) in equilibrated slice culture medium (37°C/5% CO2). Culture medium contained 78.5% Minimum Essential Medium (MEM), 15% heat-inactivated horse serum, 2% B27 supplement, 2.5% 1 M HEPES, 1.5% 0.2 M GlutaMax supplement, 0.5% 0.05 M ascorbic acid, with additional 1 mM $CaCl_2$ and 1 mM $MgSO_4$ (all from Thermo Fisher Scientific; Waltham, MA). Medium was refreshed every 3–4 days. Cultures were transfected at 4–7 days in vitro (DIV) by single-cell electroporation (SCE) and recordings were performed 4–6 days after transfection.

## Conditional AMPAR knockout using P0 viral injection

Mice with floxed loci at *Gria1*, *2* and *3* genes [*Gria1*^lox/lox (RRID:IMSR_JAX:019012), *Gria2*^lox/lox (RRID:IMSR_EM:09212), *Gria3*^lox/lox (RRID:IMSR_EM:09215)] were a gift from Rolf Sprengel (MPI - Heidelberg) and were interbred to produce mice homozygous for all floxed alleles (*Gria1*^lox/lox; *Gria2*^lox/lox; *Gria3*^lox/lox, denoted *Gria1-3fl*). 0.5 μl of AAV9-hSyn-Cre-GFP (Penn Vector Core, USA) (titre - 2 × 10^12 GC/ml) was injected into each hippocampus of *Gria1-3fl* mice at postnatal day 0–1 (P0/1) using a borosilicate glass micropipette and a 5 μL syringe (Model 75, Hamilton Company; Reno, NV). Pups were anaesthetized with 4 % Isoflurane in an anesthetic induction chamber for 3–4 min and subsequently transferred to a stereotactic rig where they were subjected to intracerebral injection, with anesthetic maintained throughout the procedure. Following recovery, pups were returned to their home cage and were used at P6-8 for the preparation of organotypic slices.

## Single-cell electroporation

Organotypic slices were transfected using an adapted version of the single-cell electroporation method described in (*Rathenberg et al., 2003*). DNA plasmids were diluted to 33 ng/μL with potassium-based intracellular solution and the mixture was back-filled into borosilicate microelectrode pipettes. Slices were submerged in HEPES-based artificial cerebrospinal fluid (aCSF) containing (in mM): 140 NaCl, 3.5 KCl, 1 $MgCl_2$, 2.5 $CaCl_2$, 10 HEPES, 10 Glucose, 1 sodium pyruvate, 2 $NaHCO_3$, at pH 7.3. Plasmids were introduced into individual cells by the application of a short burst of current pulses (60 pulses at 200 Hz) while in cell-attached mode. To visualize transfected cells, pN1-EGFP (Clontech; Mountain View, CA) was routinely mixed with AMPAR-expressing plasmids at a base pair ratio of 1:7. In the CaMKII experiments, the ratio between tCaMKII-EGFP and AMPAR-expressing plasmids was 1:1.

## Dissociated hippocampal cultures

All procedures were carried out in accordance with UK Home Office regulations. Briefly, E18 Sprague Dawley rats were sacrificed, embryonic hippocampi were isolated in HEPES-buffered Hank's balanced saline solution (Thermo Fisher Scientific) and hippocampal cells were dissociated using 0.25 % trypsin (Thermo Fisher Scientific). Cells were cultured on glass coverslips (Hecht Assistent; Germany) coated with poly-L-lysine (Sigma-Aldrich; UK) and maintained in equilibrated culture medium (37°C/5% CO2) containing Neurobasal Medium, B27 supplement (0080085SA) and Gluta-Max (all from Thermo Fisher Scientific). Cultures were transfected using Lipofectamine 2000 (Thermo Fisher Scientific) at 14–16 days in vitro and used 3–6 days after transfection.

## Electrophysiology

Transfected hippocampal slice cultures were submerged in aCSF containing (in mM): 125 NaCl, 2.5 KCl, 1.25 $NaH_2PO_4$, 25 $NaHCO_3$, 10 glucose, 1 sodium pyruvate, 4 $CaCl_2$, 4 $MgCl_2$ and 0.001 SR-95531 at pH 7.3 and saturated with 95% $O_2$/5% $CO_2$. 100 µM D-APV was used to isolate AMPAR currents for mEPSC and rectification index recordings. With the exception of mEPSC recordings, 2 µM 2-chloroadenosine was added to aCSF to dampen epileptiform activity. 1 µM tetrodotoxin was included in aCSF for miniature EPSC (mEPSC) recordings. All drugs were purchased from Tocris Bioscience. 3–6 MΩ borosilicate pipettes were filled with intracellular solution containing (in mM): 135 $CH_3SO_3H$, 135 CsOH, 4 NaCl, 2 $MgCl_2$, 10 HEPES, 4 $Na_2$-ATP, 0.4 Na-GTP, 0.15 spermine, 0.6 EGTA, 0.1 $CaCl_2$, at pH 7.25. Paired recordings involved simultaneous recording from a neighboring pair of GFP positive and negative cells. EPSCs were evoked by simulation of Schaffer collaterals in the stratum radiatum of CA1 using a monopolar glass electrode, filled with aCSF. Recordings were collected using a Multiclamp 700B amplifier (Axon Instruments). Recordings during which the series resistance varied by more than 20% or exceeded 20 MΩ were discarded. mEPSC detection was conducted using a template-based search in Clampfit (Molecular Devices). Cumulative frequency plot was produced using equal numbers of events from all cells within each condition to prevent misrepresentation. Regarding interpretation of mEPSC data, it is of note that changes in mEPSC amplitude and frequency require careful interpretation due to the event detection limit. A postsynaptic increase in event amplitude will cause previously sub-threshold events to be detected, and therefore, while the *average* event amplitude will not change, this would instead be represented as an increase in mEPSC frequency.

Rectification index was calculated by recording AMPAR currents from cells held at −60, 0 and +40 mV, using the following equation:

$$RI = -\frac{(I_{+40} - I_0)}{(I_{-60} - I_0)}$$

AMPAR/NMDAR EPSCs were compared by recording synaptic currents at −60 and +40 mV. AMPAR current amplitudes were quantified as the peak current at −60 mV. NMDAR amplitudes are measured at +40 mV, 100 ms after response initiation. Paired-pulse ratio was calculated from two AMPAR currents, stimulated at an interval of 50 ms.

For LTP recordings, aCSF contained (in mM): 119 NaCl, 2.5 KCl, 1 $Na_2HPO_4$, 26 $NaHCO_3$, 4 $CaCl_2$, 4 $MgCl_2$, 11 glucose, 0.002 2-chloroadenosine and 0.01 SR-95531 and glass pipettes were filled with intracellular solution containing (in mM): 115 $CsCH_3SO_3$, 20 CsCl, 10 HEPES, 2.5 $MgCl_2$ 4 $Na_2$-ATP, 0.4 Na-GTP, 10 phosphocreatine, 0.1 spermine at pH 7.3. Slices were maintained at 25°C throughout the recordings. LTP was induced by depolarization of the cell to −10 mV while stimulating the test pathway at 2 Hz for 100 s. The control pathway did not receive input during this period.

Outside-out patches were pulled from GFP positive or negative CA1 cell bodies and patches were subjected to fast-exchange perfusion in HEPES-based aCSF (see SCE) containing 100 uM cyclothiazide, with or without 1 mM L-glutamate. In voltage-clamp mode, a 500 ms holding potential ramp from −100 mV to +100 mV was applied to patches. Recordings in the absence of glutamate were subtracted from those in the presence of glutamate and −60 mV, 0 mV and +40 mV current amplitudes were used to calculate rectification index as described above.

## Peak-scaled non-stationary fluctuation analysis

Miniature EPSC recordings, digitized at 100 kHz, were subjected to noise analysis using a custom program running in MATLAB (MathWorks) (supplied by Andrew Penn, University of Sussex; available on MATLAB File Exchange, ID: 61567; https://uk.mathworks.com/matlabcentral/fileexchange/61567-peaker-analysis-toolbox) following (*Hartveit and Veruki, 2007*) and (*Benke et al., 2001*). Briefly, events were detected using a template-based search (*Pernía-Andrade et al., 2012*), aligned by their point of steepest rise and peak scaled to account for differences in synaptic receptor number. Traces were filtered to those with a 10–90% rise time of less than 0.9 ms and subjected to visual inspection to eliminate obvious artifacts, overlapping mEPSCs or insufficient peak alignment. Correlations between peak amplitude, rise and decay times were analyzed to detect and eliminate cells with excessive electrical filtering. Following elimination of suboptimal events, only cells with at least 20 successful events were included for variance analysis. Variance vs amplitude plots were produced for binned decay phase data of mEPSCs (15 bins) and were fitted with a parabolic curve with the equation:

$$\sigma^2(I) = iI - \frac{I^2}{N} + \sigma_b^2$$

from which single-channel current (i) could be calculated, being proportional to the initial gradient of the parabolic curve. Single-channel conductance is related to current by the equation;

$$\gamma = \frac{i}{(V_m - E_{rev})}$$

where membrane potential ($V_m$) and reversal potential ($E_{rev}$) were −60 mV and 0 mV respectively.

## Anatomical imaging

To visualize dendritic spines, 1 mg/ml Lucifer Yellow was added to the intracellular solution. Cells were maintained in a whole-cell configuration for 10 min before live imaging on an inverted Leica SP8 confocal microscope in SCE extracellular solution. Z-stacks of 50 µm regions of secondary dendrite were imaged using a 63X oil-immersion objective, deconvolved (Huygens Professional), and segmented (Imaris), before manual counting of spines.

## Fluorescence recovery after photobleaching (FRAP)

Hippocampal cultures were cotransfected (1:1) with pN1-mCherry (Clontech) and SEP-GluA2 or SEP-GluA2 ΔNTD and imaged in aCSF containing (in mM): 150 NaCl, 2.5 KCl, 2 MgCl₂, 2 CaCl₂, 20 HEPES, 10 Glucose at pH 7.3 in a heated chamber at 37°C. Images were acquired on a Leica SP8 confocal microscope using a 63X objective lens at 30 s intervals. Photobleaching was achieved by repetitive xy scanning of the region of interest at high laser intensity. Fluorescence during bleaching was monitored to ensure steady state complete bleaching was achieved and bleaching parameters were constant for all samples and repetitions. Analysis was conducted using Image J (*Schneider et al., 2012*). Photobleaching due to image acquisition was corrected by normalization to non-photobleached spines or dendrites, distant to a bleached spine.

## Flow cytometry

HEK293T cells (ATCC Cat# CRL-11268, RRID:CVCL_1926, Lot 58483269: identity authenticated by STR analysis, mycoplasma negative) were co-transfected with pN1-EGFP and AMPAR constructs using Effectene (QIAGEN; Germany). Two days post-transfection, cells were washed in phosphate buffered saline (PBS) and incubated with AF647 conjugated primary antibody (anti-myc 9E10, Santa Cruz Biotechnology; Dallas TX, RRID:AB_627268) for 30 mins on ice in PBS containing 10% fetal bovine serum (FBS). Antibody was removed and cells were washed further in PBS before resuspension in PBS containing 10% FBS and 1:1000 DAPI. Flow cytometry was performed using a LSR II flow cytometer (BD; Franklin Lakes, NJ). AF647 fluorescence was quantified and represents construct surface expression. Cells either positive for DAPI fluorescence or negative for EGFP fluorescence were discarded from analysis as dead or untransfected. AF647 fluorescence of untransfected cells was measured and subtracted during quantifications of surface expression.

## Statistics and data analysis

All data are presented as Mean ± Standard Error of the Mean (SEM). With two-sample comparisons, paired or unpaired Student's t-tests are applied as appropriate. For multiple sample comparisons, One-way ANOVA with a Tukey's multiple comparisons test was used.

## Acknowledgement

The authors would like to thank Ole Paulsen, Tim Benke and Nick Barry for advice and stimulating discussions. We are greatly indebted to Andrew Penn for providing NSFA scripts, Rolf Sprengel for providing conditional *Gria* knockout mice, José Esteban for providing the tCaMKII plasmid, Jonathan Hanley for providing SEP DNA, James Krieger for the sequence conservation model and the bio-medical staff at the Laboratory of Molecular Biology and Ares facilities for their technical support and assistance. Terunaga Nakagawa, A Radu Aricescu, Ole Paulsen, Andrew Penn and members of the Greger lab are gratefully acknowledged for critical reading of the manuscript. This work was sup-ported by grants from the Medical Research Council (MC_U105174197) and BBSRC (BB/N002113/1).

## Additional information

### Funding

| Funder | Grant reference number | Author |
|---|---|---|
| Medical Research Council | MC_U105174197 | Jake F Watson<br>Hinze Ho<br>Ingo H Greger |
| Biotechnology and Biological Sciences Research Council | BB/N002113/1 | Ingo H Greger |

The funders had no role in study design, data collection and interpretation, or the decision to submit the work for publication.

### Author contributions

JFW, Conceptualization, Data curation, Investigation, Writing—original draft, Writing—review and editing; HH, Data curation, Investigation, Writing—review and editing; IHG, Conceptualization, Supervision, Funding acquisition, Writing—original draft, Writing—review and editing

### Author ORCIDs

Jake F Watson, http://orcid.org/0000-0002-8698-3823
Ingo H Greger, http://orcid.org/0000-0002-7291-2581

### Ethics

Animal experimentation: All procedures were carried out under PPL 70/8135 in accordance with UK Home Office regulations. Experiments conducted in the UK are licensed under the UK Animals (Sci-entific Procedures) Act of 1986 following local ethical approval.

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
