## [Decision Letter]

Thank you for submitting your article "Synaptic transmission and plasticity require AMPA receptor anchoring via its N-terminal domain" for consideration by *eLife*. Your article has been reviewed by three peer reviewers, and the evaluation has been overseen by a Reviewing Editor and Richard Aldrich as the Senior Editor. The following individuals involved in review of your submission have agreed to reveal their identity: Christophe Mulle (Reviewer #2) and Johannes Hell (Reviewer #3).

The reviewers have discussed the reviews with one another and the Reviewing Editor has drafted this decision to help you prepare a revised submission.

Summary:

The study describes different properties of overexpressed GluA1 or A2 homomeric receptors and N- or C-terminal mutants in synaptic transmission in organotypic slice cultures. The authors primarily used electrophysiological assays of transfected pyramidal neurons, including an elegant electrophysiological tagging technique to identify the localization and function of recombinant receptors. To study membrane trafficking of the constructs, the authors used pHluorin-tagged receptor constructs and observed their surface recovery with FRAP. Subunit specific effects of the N- and C-terminus of GluA1 and A2 were observed. In particular, the N-terminal domains (NTDs) of the AMPAR GluA1 and GluA2 subunits promote postsynaptic accumulation of AMPARs by reducing AMPAR mobility at postsynaptic sites. Moreover, the NTD of GluA1 is required for LTP.

Essential revisions:

The main concern of the referees relates to the fact that all conclusions are based on overexpression of receptors while neither the degree of overexpression nor the ratio of exogenously expressed versus endogenously expressed receptors are known. This may compromise the conclusions. One way to address this issue could be to reduce the amount of cDNA used in the transfection experiments (e.g. GluA2Q vs. GluA2Qdelta NT, or GluA1 vs. GluA1deltaNT) in order to confirm that the conclusions hold.

A second major concern relates to the mechanism by which overexpression of GluA2Q increases evoked synaptic transmission. According to the data presented, the mEPSC amplitude and frequency are not bigger compared to control, while the single channel conductance is. To maintain the mEPSC amplitude despite higher single channel conductance with GluA2Q overexpression, less AMPA receptors would be expected to be present per synapse. Is this the case or is there another explanation for these puzzling findings?

---

## [Author Response]

*Essential revisions:*

*The main concern of the referees relates to the fact that all conclusions are based on overexpression of receptors while neither the degree of overexpression nor the ratio of exogenously expressed versus endogenously expressed receptors are known. This may compromise the conclusions. One way to address this issue could be to reduce the amount of cDNA used in the transfection experiments (e.g. GluA2Q vs. GluA2Qdelta NT, or GluA1 vs. GluA1deltaNT) in order to confirm that the conclusions hold.*

As outlined in the revised text (subsection “NTD dependent anchoring maintains synaptic AMPARs in a knockout background”, first paragraph), overexpression and knockout/rescue studies contribute different insights to receptor trafficking, each with advantages and disadvantages. Overexpression has the advantage that the level of synaptic incorporation can be measured and compared between exogenous AMPAR subunits (and their mutants) by the change in rectification index (RI), caused by replacement of endogenous with exogenous receptors. Mutants with deficits in trafficking, causing them to be less well-disposed than native receptors to anchor at the synapse can be better assayed with overexpression than in a knockout background where, without such competition, deficits will not be clearly exhibited. This approach has been successfully exploited by studies from the Malinow lab (e.g. Shi et al. Cell 2001).

Although the knock-out scenario allows one to assay the exogenously expressed subunit selectively in a null background, it creates an artificial scenario in its own right: removing other subunits (for ~ 2 weeks) may result in compensatory changes, and also the differences between wild-type and mutant receptors may not be as pronounced due to the lack of competition with fully functional receptors, as previously described. To obtain a more complete picture we have both assayed the degree of overexpression in our system, and applied a knockout/rescue approach using conditional AMPAR knockout mice to compare with our overexpression data.

We have presented three additional experimental approaches in the revised manuscript:

1) We have sought to quantify the degree of overexpression and to compare the contribution of exogenously and endogenously expressed receptors by measuring the amplitude of glutamate-evoked currents from surface patches of untransfected or overexpressing cells. These data, presented in Figures 1D2 for GluA2 and 5B2for GluA1 in the revised manuscript(reproduced in Figure 9), show that all AMPAR constructs increase the patch amplitude by an equivalent amount and by approximately two-fold when compared to untransfected cells. Given that the single-channel conductance of the Q/R- unedited, (‘Q-pore’) recombinant receptors is twice that of the native, ‘R-pore’ containing receptors (see Figure 2D2 in the original submission; Swanson et al. J. Neurosci. 1997), the number of surface receptors on overexpression is not dramatically increased, and is in fact strikingly similar to native conditions.

Author response image 1.Amplitude of glutamate-evoked AMPAR currents from outside-out patches of untransfected and AMPAR construct transfected CA1 pyramidal neurons.Amplitude of currents is similar between all transfected cells and appears to be double that of untransfected cells.**DOI:**
http://dx.doi.org/10.7554/eLife.23024.017

2) We have assessed whether association of TARP auxiliary subunits with exogenous receptors is reduced as a result of overexpression (i.e. whether TARPs are limiting). AMPAR auxiliary subunits are critical mediators of receptor trafficking (most prominently the TARPs), both in receptor delivery to the cell surface, and as a synaptic anchoring mechanism by binding to PSD-95 (see Chen et al. Nature2000; Schnell et al., PNAS2002). Hence, we assayed the level of auxiliary subunit association by recombinant receptors to understand if overexpressed receptors were less associated with auxiliary subunits than native receptors, and secondly, to determine if mutant AMPAR constructs show differential association with auxiliary subunits, which could explain the synaptic phenotypes that we observe.

As AMPAR association with type-1a TARP auxiliary subunits elevates kainate efficacy, the kainate/glutamate ratio provides a reliable measure for the presence of these predominant auxiliary subunits (e.g. Shi et al. Neuron2009). We have used this assay on excised membrane patches and find that the KA/Glu ratio was no different either between wild-type and ΔNTD receptors, between GluA subunits or on comparison to native receptors, showing that auxiliary subunit association is comparable in all conditions (revised manuscript Figure 1 and Figure 5, reproduced in Figure 10). This result confirms that 1) our overexpression does not appear to be at an extreme, or non-physiological level, as full auxiliary subunit association is observed, and 2) reduced auxiliary subunit interactions cannot account for the NTD dependent synaptic anchoring that we describe. These data are also in accordance with an earlier study by Kessels and Malinow (Kessels et al., Nat. Neurosci. 2009), who overexpressed GluA1 and GluA2 virally and demonstrated that TARPs are not limiting for dendritic AMPARs. Our new data are explained in the ‘Results’ section of the revised manuscript.

Author response image 2.KA/Glu ratio of AMPAR currents from outside-out patches is unchanged on overexpression of exogenous AMPAR constructs.**DOI:**
http://dx.doi.org/10.7554/eLife.23024.018

3) We have investigated NTD-dependent AMPAR anchoring using a rescue strategy in AMPAR conditional knockout neurons, to study synaptic anchoring using an alternative approach. Specifically, we have rescued conditional AMPAR knockout neurons (in slices from *Gria1-3 floxed* mice), with recombinant GluA1 ± NTD and GluA2Q ± NTD subunits, comparing synaptic current amplitudes to native neurons (not experiencing knockout or rescue) in paired recordings (see Figure 6 of the revised manuscript). Using this approach, we also see a clear dependence of GluA2 on its NTD for synaptic anchoring, deletion of which reduces current rescue by approximately 50% . While GluA1 ΔNTD is able to rescue synaptic currents in the absence of native receptors (compare Figure 6 with Figure 5 of the revised manuscript), rescue was similarly impaired on comparison to wild-type GluA1. Interestingly, GluA2 rescue was substantially higher than GluA1, mirroring the differences in RI seen on overexpression.

Taken together these additional data sets demonstrate that our overexpression approach is neither excessive, nor dramatically non-physiological and we believe that the additional insights we have gained adds considerable strength to our original conclusions.

*A second major concern relates to the mechanism by which overexpression of GluA2Q increases evoked synaptic transmission. According to the data presented, the mEPSC amplitude and frequency are not bigger compared to control, while the single channel conductance is. To maintain the mEPSC amplitude despite higher single channel conductance with GluA2Q overexpression, less AMPA receptors would be expected to be present per synapse. Is this the case or is there another explanation for these puzzling findings?*

We agree that this is an important point, however it is difficult to draw such conclusions from mEPSC data. Regarding the mechanism for increased evoked transmission, overexpression of GluA2Q substantially increases the EPSC amplitude (Figure 1 of the original manuscript), which is likely to be predominantly mediated by the change in synaptic channel conductance as has been stated in the revised manuscript (Discussion, sixth paragraph).

However, mEPSC recordings are more difficult to decipher due to the event detection limit. Many events occur, which are not detected as they are below the detection limit. Actual increases in event amplitude will allow previously sub-threshold events to be detected, increasing the mEPSC frequency, yet the average amplitude will likely be unaffected due to the contribution of a greater number of small, previously undetected events to the average value. Decreases in event amplitude can similarly affect mEPSC frequency. This caveat was stated in the Methods section of the original manuscript but has now been more directly emphasized the revised manuscript (subsection “GluA2 ΔNTD reduces spontaneous transmission”, second paragraph and subsection “Electrophysiology”, first paragraph). Such interplay between mEPSC amplitude and frequency has been highlighted previously by multiple laboratories (Lu et al. Neuron 2009; Rumbaugh et al. PNAS 2006).

While the difference in mEPSC frequency between untransfected and GluA2Q expressing cells is not statistically significant because these data are highly variable (untransfected frequency ranges from 0.14 to 1.08 Hz), both the apparent elevation in mEPSC frequency (Figure 2 of the original manuscript) by GluA2Q and the increase in amplitude of large mEPSCs evident in Figure 2 (original manuscript)are indicative of an increase in event amplitude that will not be represented by the mean event amplitude (Figure 2 of the original manuscript), because of the limitations described above.

We do not feel that mEPSC data is sufficiently informative to speculate on such specific details of synaptic AMPAR content, given that the interplay between event amplitude and frequency, with evoked EPSC data being more reliably interpreted for this purpose. As previously stated, evoked EPSCs shows a clear increase in amplitude, which we believe convincingly argues against a drastic decrease in synaptic receptor numbers. For the reasons detailed above we assert caution not to over-interpret mEPSC data, however our mEPSC dataset unambiguously shows differences between GluA2Q and GluA2Q ΔNTD for both amplitude and frequency, strongly supporting our conclusions regarding the requirement for the NTD in synaptic anchoring.